# Recent Developments in Wireless Soil Moisture Sensing to Support Scientific Research and Agricultural Management

**DOI:** 10.3390/s22249792

**Published:** 2022-12-13

**Authors:** Heye Reemt Bogena, Ansgar Weuthen, Johan Alexander Huisman

**Affiliations:** Institute of Bio- and Geosciences, Agrosphere Institute (IBG-3), Forschungszentrum Jülich GmbH, 52425 Jülich, Germany

**Keywords:** soil moisture sensing, wireless communication technology, field and catchment scale, sensor validation, remote sensing, hydrological modelling, cyberinfrastructure

## Abstract

In recent years, wireless sensor network (WSN) technology has emerged as an important technique for wireless sensing of soil moisture from the field to the catchment scale. This review paper presents the current status of wireless sensor network (WSN) technology for distributed, near real-time sensing of soil moisture to investigate seasonal and event dynamics of soil moisture patterns. It is also discussed how WSN measurements of soil measurements contribute to the validation and downscaling of satellite data and non-invasive geophysical instruments as well as the validation of distributed hydrological models. Finally, future perspectives for WSN measurements of soil moisture are highlighted, which includes the improved integration of real-time WSN measurements with other information sources using the latest wireless communication techniques and cyberinfrastructures.

## 1. Introduction

Knowledge of soil moisture distribution in landscapes is essential, as soil moisture represents a key variable in many pedological, geomorphological, hydrological, climatological, environmental, and ecohydrological processes. Soil moisture plays an important role in the hydrologic cycle by partitioning precipitation into runoff and infiltration [1] and by controlling hydrological fluxes, such as interflow [2] and groundwater recharge [3]. Soil moisture also has a strong influence on energy fluxes between the land surface and the atmosphere, as well as on evaporation and transpiration fluxes, which makes it a key variable in the Earth’s climate system [4]. Recently, Humphrey et al. [5] showed that interannual fluctuations in terrestrial water storage strongly affect the terrestrial carbon sink and highlighted the importance of the interactions between the water and carbon cycle.

Soil moisture is highly variable in spatial and temporal scales and is controlled by complex, interrelated environmental factors (e.g., soil properties, topography, global radiation, precipitation, and vegetation) which can also change spatially and temporally [6]. Because soil moisture influences hydrologic processes, including evaporation and drainage, in a nonlinear manner [7], a better understanding of the dynamics of spatial soil moisture variability also provides valuable insights into the functioning of hydrological systems [8,9,10,11,12,13]. To this end, great efforts have been undertaken during the last decades to determine soil moisture pattern dynamics and to understand their controlling factors [14,15]. Soil moisture datasets have been acquired at different spatial scales, including datasets with point measurements from the plot to the small catchment scale, e.g., [12,16], as well as remote-sensing-based datasets, e.g., [17,18], and modeled soil moisture datasets, e.g., [19,20], at larger scales.

In their seminal article, Blöschl and Sivapalan [21] proposed the “scale triplet” to describe different spatial scales relevant for soil moisture data. The scale triplet consists of the support, spacing, and extent of soil moisture data. The support relates to the integration volume of the measurement procedure, the spacing relates to the distance between individual measurements, and the extent relates to the area covered by the available measurement data (e.g., area of the measurement network). Similarly, the scale triplet for soil moisture time series can be defined by: (1) the measurement integration time, e.g., continuous, intermittent, and day and night; (2) the measurement frequency; and (3) the measurement period.

The complex nature of soil moisture requires a multidisciplinary approach to investigation that encompasses different subfields of physical geography [22]. Figure 1 illustrates the linkages between soil moisture and weather/climate, geomorphology/hydrology, and biogeography for different soil moisture conditions. For instance, soil moisture particularly strongly affects weather and climate during dry and wet conditions through changes in albedo and the partitioning between latent and sensible heat. Similarly, soil moisture strongly influences hydrological processes during dry and wet conditions, e.g., occurrence of low flow and flood events. Geomorphological processes exert a strong control on the water-holding properties of soil and, thus, on soil moisture patterns during dry and intermediate conditions. Soil moisture affects biogeography during wet (e.g., root development), intermediate, and dry conditions, whereas the influence of biogeography on soil moisture is strongest during intermediate conditions. A more detailed discussion on the various interactions of soil-moisture-related processes is given in Legates et al. [22].

Since the first attempts to determine soil moisture variability at the field scale, e.g., [23], considerable advances have been made in the measurement of soil moisture. Since the second half of the last century, soil moisture has become an important topic in environmental research. An analysis based on the Web of Science Core Collection database revealed that more than 65,000 research articles related to soil moisture topics have been published since 1961, with a sharp increase in the number of annual publications around 1990 and an exponential growth in publications since then (Figure 2). For this analysis, articles were identified using a Web of Knowledge search query: Soil moisture: Topic = (“soil moisture” OR “soil water content”; Soil sensor: Topic = (“soil” AND “sensor”); Sensor network: Topic = (“soil*” AND “sensor*” AND “network*”); Cosmic-ray neutron sensor: Topic = (“cosmic*” OR “cosmos*” AND “soil moisture” OR “soil water content”); Document types: (ARTICLE).

The increased interest of the scientific community in soil moisture is strongly linked to the technological progress in soil moisture sensors (Figure 2). For instance, the development of the time domain reflectometry (TDR) approach for soil moisture determination in the 1970s was a major breakthrough in soil hydrology [24,25]. The seminal work of Topp et al. [25] demonstrated the strong and stable relationship between the soil permittivity and soil moisture, which enabled the use of TDR instruments for the relatively easy and continuous monitoring of soil moisture dynamics. New developments in soil moisture measurement techniques during the last two decades, for instance microwave remote sensing [17,18,26] and wireless sensor networks making use of low-cost soil moisture sensors [14,16], have increased the availability of soil moisture information to a large degree. In addition, new soil-moisture-monitoring techniques that allow continuous noninvasive and contactless measurements of soil moisture dynamics from the field to the catchment scale are emerging [27], such as cosmic-ray neutron sensors (Figure 2).

The first studies on the spatial variability of soil moisture relied on TDR measurements at a few selected time points and on the topsoil [28]. In addition, continuous TDR measurements were restricted to the plot scale due to limited cable lengths (max. 20 m). However, a limited number of time points of spatial soil moisture measurements may not be sufficient to capture the temporal dynamics of spatial soil moisture variability on a larger scale [7], to analyze spatial dependence [29], and to determine temporal dynamics of spatial soil moisture variability after precipitation events [30,31].

With the introduction of wireless sensor networks (WSNs) in soil moisture sensing, continuous three-dimensional soil moisture data with high temporal resolution, sufficient spatial coverage, and vertical resolution above the root zone have become available, which significantly improved our process understanding of hydrological systems at the catchment scale, e.g., [14,32,33]. Such data are especially valuable for the analysis of lateral water fluxes at the hillslope and catchment scale, for example to evaluate hydrological models [19,20,34] and to improve the prediction of hydrological fluxes using data assimilation [28]. Therefore, WSNs also play an important role in the development of terrestrial environmental observatories [35,36]. Recently, Narrow Band Internet of Things (NB-IoT) communication technology has revolutionized the real-time measurement of soil moisture. Therefore, we feel it is important to provide an overview of the latest developments in wireless measurement of soil moisture in support of scientific research and agricultural management.

This review paper presents recent developments in WSN technology for distributed and real-time sensing of soil moisture from the field to the catchment scale and its application in fundamental and applied environmental research. In addition, we also provide an overview of appropriate sensor techniques used in soil moisture WSN applications, as well as a wide range of soil moisture WSN applications, including examples from hydrological research, remote sensing and geophysics, hydrological modeling, and agricultural management. In this way, interested users are given a complete overview of the potential of soil moisture WSN without getting lost in technical details.

The review paper is organized as follows: First, the most common WSN techniques that can potentially be used for soil moisture measurements are presented. This is followed by an overview of common types of soil moisture sensors used for WSN applications. Next an overview of applications of wireless soil moisture measurement in hydrology and related fields is given, which includes both basic as well as applied research such as irrigation management. The paper concludes with a summary and future perspectives.

## 2. Wireless Sensor Network Technology

WSN technology enables distributed sensing through efficient data communication between a multitude of environmental sensors [14,37,38]. WSN is still a relatively new area of research, but the communication technology used for low-cost, low-power wireless networks has advanced greatly in recent decades [39]. A well-designed WSN is in principle infinitely expandable and data acquisition can be dynamically adapted to external influences. For example, a soil moisture WSN can be connected to a rain gauge, and the frequency of sampling can be automatically increased during rain events to allow the study of highly dynamic processes with efficient energy use. An important advantage of long-term sensor network measurements compared with conventional measurement campaigns with mobile sensors, such as, e.g., [31], is that the soil moisture probes can remain continuously in the soil. In this way, comparisons among different points in time are not affected by different measurement locations or changes in probe properties (e.g., due to different calibration quality).

Low-power wide-area technology (LPWA) is an umbrella term for various technologies that enable WSN communication at a relatively low cost and with reduced power consumption. Many LPWA technologies have emerged in both licensed and unlicensed markets, e.g., ZigBee, SigFox, Long Range (LoRa), and Narrow Band (NB)-IoT. In the following section, the most recent and widely used LPWA technologies, i.e., ZigBee, LoRa and NB-IoT, are discussed in more detail.

### 2.1. ZigBee Wireless Sensor Network Technologies

Among the first proprietary standards for low-cost mesh networks used for WSNs was ZigBee [40]. ZigBee is a set of high-level communication protocols that utilize 2.4 GHz low-power radio modules based on the IEEE 802.15.4 LPWA standard [41,42]. More recently, the ZigBee technology has been further developed into ZigBeePro [43] and JenNet [44]. Each component of a ZigBee based WSN has a radio module to enable wireless communication. The ZigBee radio modules have several software interfaces that connect the hardware devices (physical layer and peripherals) to the user application. The user has the possibility to control the sensor network and manage the communication between the devices by means of the application support layer (APS) and the application programming interface (API). The routing of data within the network and data transmission is handled by the media access control layer (MAC). This is based on the IEEE 802.15.4 standard and is located on the physical layer (PHY). The PHY layer includes the transceiver as well as the sensors and the power source [41]. Finally, the user has the possibility to realize advanced functions (e.g., logging function, sensor driver, etc.) by developing a special user software that configures the sensor network.

One example of a ZigBee-based soil moisture WSN is SoilNet [14,45], which was developed at the Forschungszentrum Jülich using the proprietary license free protocol stack JenNet developed by Jennic Ltd., South Yorkshire, UK [44]. JenNet uses the unlicensed 2.4 GHz band and supports star, tree, and linear topologies. In the case of a tree topology, JenNet can support WSN with up to 250 nodes and in the case of a linear topology, even up to 1000 nodes. JenNet transmission distances are limited to less than 100 m in the case of underground WSN. That is why SoilNet uses a hybrid WSN method consisting of a mixture of underground terminals, each wired to multiple ground sensors and above-ground router devices. This allows significantly greater transmission ranges of up to several 100 m and enables SoilNet to cover whole catchment areas (Figure 3).

### 2.2. LoRa Wireless Sensor Network Technology

Due to the use of 2.4 GHz low-power radio modules, the range of wireless communication between ZigBee nodes is limited to a few kilometres. Therefore, more recently, the LoRa (Long Range) communication technology has been introduced for long-range, low-power, low-bit-rate wireless communication, enabling larger WSN coverage with power consumption similar to ZigBee by using chirp spread spectrum (CSS) modulation technology [46]. This modulation technique maintains the same low power characteristics as standard radio modulation but significantly increases the communication range because it is more robust to interference.

LoRa consists of the network protocol (LoRaWAN) and the associated hardware components, such as radio module and antennas and is optimized for battery-powered devices [46]. LoRaWAN uses star topologies with three different types of devices: end devices (also called LoRa nodes) that can host a set of environmental sensors, a LoRa gateway, and a LoRa network server [47]. The basic structure of a LoRaWAN wireless network is presented in Figure 4. The LoRa network server is the top of the network tree and stores information about the network, initiates the wireless links within the network, and can connect to a database server (Figure 4). The LoRa gateway acts as a relay station that passes data from the sensor devices to the LoRa server, where it can be processed by the LoRa application software. The LoRa end devices are the environmental sensors, which should have just enough functionality to communicate with the gateway. This allows the LoRa end devices to be asleep a significant amount of the time to save energy.

Since LoRa networks are specifically designed to be applied to larger areas [39], it is important for the planning and optimization of LoRa networks to know the coverage probabilities depending on the distance between the LoRa transmitting and receiving stations. For this reason, the radio technology of LoRa and the calculation of the potential radio link distance is discussed in more detail in the following. At the heart of LoRa is a proprietary chirp spread spectrum (CSS) modulation technique [48]. For binary chirp modulation, the data passes through a chirp modulator that maps each bit block to 1 of 2 waveforms. The chirped LoRa signal can be described by:(1)$$s\left(t\right)=\sqrt{\frac{2E_{s}}{T_{s}}}cos\left[2\pi f_{c}t\pm \pi \left(u\left(\frac{t}{T_{s}}\right)-w{\left(\frac{t}{T_{s}}\right)}^{2}\right)\right]$$
where *E_s_* is the energy of *s(t)* in the symbol duration *T_s_*, *f_c_* is the carrier frequency, and the parameters *u* and *w* are the peak-to-peak frequency deviation and the sweep width, respectively, both normalized by the symbol rate. LoRa supports variable data rates, enabling the trade-off between throughput, range, robustness, and power consumption while maintaining bandwidth. The LoRa server manages these aspects by regulating the bandwidth *BW* and the so-called spreading factor *SF* that determines the length of the chirp symbol. The time-on-air of a transmission increases exponentially with *SF*, extending the communication range between gateway and end devices. The LoRa protocol has six *SFs* (7–12, Table 1). The lower *SFs* provide higher data rates but shorter communication distance, while the higher *SFs* provide lower data rates but higher transmission stability. A signal failure in the uplink can occur at the gateway if the received signal-to-noise ratio (SNR) is below an *SF*-specific threshold value (*q_SF_*, Table 1).

The expected communication performance of the LoRa signal transmission technique can be estimated for a single end-device using the following considerations. Following the Friis’ transmission equation, the path loss *g* can be calculated as a function of distance between sender and receiver *d*:(2)$$g\left(d\right)=\frac{\lambda }{{\left(4\pi d\right)}^{\eta }}$$
where *λ* is the carrier frequency wave length, and *η* is the path loss exponent (*η* ≥ 2), usually taken to be equal to 2.7 in suburban environments. In addition, path loss due to shadowing effects can be approximated with noise variance assuming white Gaussian noise with zero mean:(3)$$\sigma^{2}=-174+NF+10log_{10}\left(BW\right)$$
where *NF* is the receiver noise and assumed to have a value of 6 dB, and *BW* is the bandwidth of *s(t)*, assumed to be 125 kHz in this case. Finally, the coverage probability *CP*, defined as the probability that the signal-to-noise ratio *SNR* is equal to or larger than the threshold value *q_SF_*, can be obtained as
(4)$$CP=exp\left(-\frac{\sigma^{2}\cdot q_{SF}}{Ρ\cdot g\left(d\right)}\right)$$
in which *P* is the transmitted power, here assumed to be 14 dBm (25.12 mW).

Figure 5 presents the coverage probability as a function of distance obtained with values of *q_SF_* from Table 1 and Equation (4). The following conclusions can be drawn from the calculations. In the case of a path loss exponent of 2.5, radio links can cover distances from 6.5 to 23 km, assuming that 90% coverage is sufficient for reliable transmission. When the path loss exponent increases to 2.7, the possible distance of the radio links decreases to 2.7 to 9 km at 90% coverage. This demonstrates the strong sensitivity of the communication distance to the path loss exponent and suggests that this value should be chosen with care. To check the appropriateness of the path loss exponent for a given area, the theoretical results should be compared with transmission performance measurements with mobile LoRa receiver and gateway devices at different distances.

Recently, a number of studies have developed and deployed low-cost LoRa-based WSNs for long-distance soil moisture monitoring to test their usability for smart agriculture [49,50]. In addition, Wang et al. [51] developed and successfully tested a LoRa-based WSN to investigate the control of soil moisture on greenhouse gas emissions in a wetland. The results of the WSN experiment by Rachmani and Zulkifli [50] showed that LoRa achieves maximum performance when the communication range is below 700 m, with average values of RSSI (received signal strength), SNR (signal-to-noise ratio) and PDR (packet–delivery ratio) of −120 dBm, 1 dB, and 40%, respectively, for the given conditions in a star fruit plantation. This indicates that local conditions, such as dense vegetation surrounding the LoRa transmitters, can have a strong negative impact on the expected radio link distances of LoRa-based WSNs.

### 2.3. Narrowband Internet of Things (NB-IoT)

The commercial narrowband Internet of Things (NB-IoT) is a cellular LPWAN (Low Power Wide Area Network) that is becoming increasingly important as an alternative to ad-hoc networks such as LoRa, especially due to the significantly lower transmission costs compared with broadband mobile radio standards [52]. NB-IoT has the same advantages as LoRa (i.e., low data rate, low power consumption and low bandwidth) and avoids the need to maintain an ad-hoc sensor network infrastructure (e.g., gateways, network server, etc.) because of the high coverage provided by cellular networks. NB-IoT has been integrated into the LTE (long-term evolution) broadband radio standard by the 3rd Generation Partnership Project (3GPP), and commercial marketing has been underway since 2018 [53]. To reduce device costs and minimize battery consumption, NB-IoT is kept as simple as possible by omitting many of the features of LTE [53]. NB-IoT also uses the licensed LTE frequency bands, and the NB-IoT core network relies on the evolved packet system (EPS) to select the best path for control and payload packets for uplink and downlink data [54]. The cell access procedure of an NB-IoT device is also similar to that of LTE. Data are transmitted via a serving gateway to a network gateway and forwarded to the application server via radio bearers that use the existing air interface and backbone (i.e., the Internet’s background network) of LTE. The use of NB-IoT is limited to 4G/LTE base stations, and the coverage should typically not be less than 23 dB for proper functioning [54]. For this reason, NB-IoT is often less suitable for rural areas than LoRaWAN.

NB-IoT already has almost nationwide coverage in China as do several North American and European countries through the major mobile phone connection providers. For instance, Deutsche Telekom has signed its first NB-IoT roaming agreements with several European operator partners, offering roaming services in 18 countries in Europe [55]. Figure 6 shows that Deutsche Telekom alone currently offers NB-IoT coverage for approximately 90% of the areas in Germany. The areas with poor NB-IoT coverage are located mainly at high altitudes and in deep valleys. In addition to this general availability of NB-IoT, local obstacles such as buildings or trees can limit the wireless connection between an NB-IoT device and the base stations, especially if the radio antenna is close to the ground, which is often the case with soil moisture sensors. However, with the continuous densification of the NB-IoT network and the increasing number of providers, these problems are continually becoming increasingly less frequent.

It has been reported that the communication delay of NB-IoT technology is generally greater than the delay of LoRa due to the complex air interface scheduling process, which also requires iterations [56]. This can have a negative effect on the maintenance intervals of IoT devices because a longer transmission time increases power consumption. On the other hand, it has been shown that NB-IoT provides better transmission performance than LoRaWAN in underwater and underground environments [57].

The development of sensor devices with communication capabilities, such as NB-IoT, has been recognized as an essential component of smart agriculture [58]. Recently, several NB-IoT-based smart water management platforms for irrigation have been proposed and implemented [59,60,61,62]. With the increasing further expansion of the worldwide spatial coverage of NB-IoT, it is anticipated that smart agriculture will also become a feasible cost-effective technical solution for smaller farms [63].

NB-IoT networks can also support public participation of non-scientists in scientific projects, e.g., through community-based data collection with the help of large numbers of volunteers. For example, the ongoing CurieuzeNeuzen project [64] involves up to 5000 citizen scientists, who installed a NB-IoT soil moisture sensor in their garden, schoolyard, park, or private property. Using NB-IoT, the measurement data from these sensors are transmitted to a database at the University of Antwerp, making the data available to the involved (citizen) scientists in real time.

## 3. Soil Moisture Sensors for Wireless Network Applications

Due to the large number of soil moisture measurements within a WSN, the reading of the sensor signal should be simple and unambiguous [14]. In addition, soil moisture sensors should be as affordable as possible to ensure that the number of sensor nodes is as high as possible [65]. Since capacitance and time-domain transmission (TDT) sensors meet these criteria best, these sensor technologies have been used most often in the development of WSNs for soil moisture (Table 2).

Low-cost soil moisture sensors are often supplied with a factory calibration that does not achieve the maximum possible measurement accuracy of the sensor. For example, Dominquez-Nino et al. [67] showed a significant overestimation of soil moisture for a capacitive sensor when the factory calibration was used. In addition, low-cost sensors typically show considerable sensor-to-sensor variability [67], which may also decrease the measurement accuracy of soil moisture sensor networks. One possible solution would be a direct calibration between sensor response and soil moisture for every sensor [79]. However, in the case of sensor network applications several hundreds of sensors are often used [16,74,80], which makes direct sensor calibration impracticable. Alternatively, a two-step calibration procedure can be applied [62]. In a first step, the relationship between sensor response and permittivity for each sensor of a wireless network is determined [81]. For this, media with well-known dielectric properties, such as air, glass beads, or dielectric liquids, such as 2-isopropoxyethanol [82] or 1,4-dioxane [83], are often used. The use of liquids is desirable to avoid air gaps and density variations and to enable a quick calibration of multiple sensors. In the second step, the dielectric permittivity is related to soil water content using empirical or semi-empirical models [25,84]. For more accurate soil moisture measurements, a site-specific calibration accounting for soil textural variation can also be performed on a limited number of samples using TDR measurements [16,66]. A more detailed discussion on the two-step calibration procedure is given in Dominquez-Nino et al. [67].

To enable efficient and economical use of limited water resources in crop irrigation, so-called soil moisture profile sensors (SMPS) for soil moisture measurement in the entire root zone are becoming increasingly popular [85,86]. SMPS are also based on the electromagnetic method and, thus, have great potential for WSN-based climate-smart agriculture due to their ease of use and simultaneous measurement at different depths [87,88].

## 4. Wireless Soil Moisture Sensing Applications

### 4.1. Applications in Hydrological Research

WSNs have already been successfully deployed in various locations around the world, providing a large number of soil moisture measurements (see references in Table 2). For instance, the SoilNet WSN installed in a headwater catchment has continuously operated since 2009 and has already produced more than 50 million soil moisture measurements [80]. Most WSN studies of soil moisture were focused on the investigation of hydrological processes at the field and catchment scale [14,16,66,80]. For instance, Bogena et al. [14] analyzed data from a SoilNet WSN implemented in a headwater catchment and found that soil moisture variability in forested mountainous regions can be remarkably high during intermediate wetness conditions. Similar results were found by other studies in mountainous regions using campaign-style soil moisture measurements with handheld sensors [31,89]. However, the SoilNet data showed less scatter than such discontinuous soil moisture measurements, indicating that WSN data provide more detailed insights into the hydrological processes generating soil moisture patterns [66,90,91]. Bogena et al. [14] also showed that soil moisture variability strongly decreased with depth, indicating that the factors controlling prolonged travel time decrease the spatial variability of soil moisture. In addition, topographic attributes had the strongest correlation with soil moisture during dry conditions, suggesting that the control of topography on soil moisture patterns depends on water storage in the soil. Finally, it could be shown that interpolation of the densely sampled point data from SoilNet allowed the capture of the key patterns of soil moisture variation at the catchment scale [14].

Using a longer time series of SoilNet data, Rosenbaum et al. [66] identified hysteresis in the relationship between spatial variability and mean soil moisture at the event and seasonal time scales and as a function of the mean soil moisture and precipitation conditions. Soil moisture data from a WSN was also used to analyze the temporal stability of soil moisture, and it was found that both soil moisture and saturation degree showed temporally stable characteristics that were correlated with the spatial variation in hydraulic parameters [86]. In a subsequent study, Qu et al. [87] developed a new closed-form expression to predict local variability of soil moisture and to estimate the spatial variability of hydraulic properties on the basis of WSN data from several sites in Germany and China.

Soil moisture data from WSN has also proven to be useful for water balance analysis at the catchment scale. For example, using an empirical orthogonal functional analysis, Graf et al. [10] were able to show that two underlying soil moisture patterns explain the overall soil water storage variability in a headwater catchment. Using the same WSN data, Wiekenkamp et al. [12] compared the spatiotemporal distribution of soil moisture before and after partial deforestation and found that soil moisture in the deforested area was significantly higher compared with the forested part, especially during the summer period. This, in turn, caused an increase in the frequency of high discharge in the first year after the deforestation. In another study, Wiekenkamp et al. [11] used soil moisture sensor response times from a WSN to investigate controls on preferential flow at the catchment scale. They found that the spatial occurrence of preferential flow showed no obvious relationship with spatial attributes, such as topographical and soil physicochemical parameters, but was governed mainly by small-scale soil and biological features and local processes. Metzger et al. [76] investigated the influence of throughfall variability on soil moisture variability in a forest stand using data from a dense soil moisture WSN and found that soil hydraulic properties were the dominant drivers for spatial soil moisture patterns.

Recently, WSNs have contributed to achieving advances in real-time hydrological monitoring [92]. For instance, WSNs have been used to develop real-time flood monitoring systems that combine soil moisture sensors with hydrometeorological sensors to provide important information for the efficient management of flood events [93,94]. For example, a ZigBee-based WSN was used for an early flood monitoring and warning system to provide flood warning. The WSN system informs the public of the water level even without network connection in their mobile application and helps them to reach a safe place in a timely manner [95]. Furthermore, real-time soil moisture WSNs have been used to develop local early warning systems for drought that combine soil moisture data from a WSN with hydrologic model data and locally observed precipitation [96].

Finally, Rabbel et al. [97] used a combination of dendrochronological data and soil moisture information from a WSN to identify climate-growth signals of forest trees at the catchment scale. They found that water availability was the dominant growth factor and that the drier sites at the hillslopes showed stronger climate-growth reactions compared with sites located in the riparian zone with less variable water availability. This promising result indicates that the detailed data from soil moisture WSNs are also useful for obtaining a deeper understanding of species-specific growth limitations.

### 4.2. Applications in Remote Sensing and Geophysics

In the past decade, soil moisture WSNs contributed to several validation activities of satellite missions such as the SMAP (Soil Moisture Active and Passive) or SENTINEL missions by providing reference data [18,98]. In addition, soil moisture data from WSNs were used for the calibration and validation of simultaneous passive/active microwave airborne campaigns in the Rur catchment [99,100] to analyze various radiometer–radar fusion methods for retrieving improved soil moisture data products [101] and to develop methods for downscaling SMAP radiometer data [17].

Soil moisture WSNs have also been used to support unmanned aerial vehicle (UAV) applications for soil moisture mapping [102]. For instance, Akbar et al. [103] showed that a combined UAV and WSN instrumentation can be used to better capture the soil moisture of a certain area by optimizing the UAV pathway planning. Other studies combined a UAV platform with a WSN to monitor, in real time, important parameters that characterize the growth conditions [104] and the thermal stress in vineyards [105].

Soil moisture data from WSNs were also used to validate field-scale soil moisture measurements with non-invasive geophysical instruments. For instance, soil moisture data from SoilNet have been used to test novel cosmic-ray neutron sensors (CRNS) that enable non-invasive soil moisture measurements at the field scale [72,106,107,108]. On the other hand, soil moisture data from SoilNet were used to demonstrate the strong influence of biomass on CRNS measurements [109]. In another study, Altdorff et al. [110] used SoilNet data to investigate the potential of mapping soil moisture with electromagnetic induction (EMI) measurements in a forested headwater catchment. They found limitations for the use of EMI to monitor spatiotemporal soil moisture changes due to low electrical conductivity of the forest soils and the spatiotemporal variability of the pore water electrical conductivity.

### 4.3. Applications in Model Validation

Soil moisture data from WSNs have been used extensively for the validation of distributed land surface models and rainfall runoff models. For instance, the soil moisture data provided by SoilNet was used to validate several distributed hydrological models, such as HydroGeoSphere [111,112], MIKE-SHE [19,20], and TerrSysMP [37,113]. For example, Fang et al. [19] used WSN soil moisture data to validate high-resolution long-term 3D simulations of a headwater catchment with the hydrological model ParFlow-CLM using empirical orthogonal function and wavelet coherence methods. Their results revealed that information on soil porosity heterogeneity effectively improves estimates of soil moisture patterns and that wet and dry seasons have a significant effect on temporal correlation between observed and simulated soil moisture. In another study, Koch et al. [20] compared three distributed hydrological models for their ability to simulate soil moisture patterns at the catchment scale and were able to analyze the spatial error correlation between these models. More recently, long-term soil moisture data from several WSN in Germany have been used to validate an operational model-based drought monitor [114].

### 4.4. Applications in Agricultural Management

Globally, one-quarter of cropland already needs to be irrigated, requiring more than 70% of human consumption of blue water [115]. Due to the increase in drought events in many regions of the world, efficient irrigation management is essential to optimize agricultural production and avoid overuse of water resources [67]. To this end, real-time monitoring of soil moisture is essential to optimize the amount and timing of irrigation. For instance, Vellidis et al. [116] developed a closed loop irrigation system where inputs from a soil moisture WSN determine irrigation timing and amounts in real-time for a field cropped with cotton. It was shown that the use of WSN to measure soil moisture led to water savings of up to 70% in irrigation compared with conventional irrigation scheduling based on calculated evapotranspiration [117,118]. However, there are also weaknesses in WSN-based irrigation scheduling. For instance, the electromagnetic sensors typically used for soil moisture monitoring have a relatively small probing volume, which typically does not capture the field average soil moisture content. The spatial variability of soil properties within fields and between farms must be considered [90]. Therefore, it has been recommended that a minimum of three WSN nodes with multi-depth soil moisture sensors are installed to cover the root zone in each homogenous irrigation sector to represent the total available water storage [117]. From the economic point of view of the farmer, the investments for the acquisition of the WSN equipment, the maintenance, and extension costs have to be taken into account. In addition, the maintenance of the WSN requires qualified personnel [118]. Moreover, WSNs typically provide massive data amounts, so efforts are needed to translate data into user-friendly information [119]. An example where this has been implemented is the ADAPTER project, which involves the development and delivery of WSN-based information products to farmers in the form of user-friendly analyses and data products to support climate resilient agriculture [120]. Here, the WSN data is used for data assimilation in a soil hydrological model to derive user-tailored predictions such as plant available soil moisture with a lead-time of 10 days.

## 5. Summary and Future Perspectives

Soil moisture WSNs have evolved significantly in terms of technology in the past two decades and can now cover larger areas with sub-gigahertz communication technologies, such as the LoRa technology, to provide detailed spatiotemporal soil moisture data at the catchment scale. In the past years, WSN data are increasingly used to support the analysis of soil hydrological processes from the field to catchment scale and the validation of remote sensing products, geophysical sensors, and process-based models. In addition, WSNs are now used for real-time management purposes, e.g., the optimization of crop yields through efficient irrigation based on soil moisture information.

Recent advances in sensor techniques allow for continuous non-invasive soil moisture measurements that integrate over scales beyond the traditional point measurement [27]. For instance, cosmic-ray neutron sensors (CRNS) provide non-invasive soil moisture estimates at the field scale with an effective radius of 130 to 240 m and a penetration depth of up to 80 cm [95]. Recently, data from a network of CRNS stations in Europe has been published [121], but a large proportion of the stations are not equipped with data transmission. Equipping all CRNS stations with WSN technology would allow the soil moisture data to be available in near real time, e.g., for use in improving flood models within the European Flood Awareness System [122].

Much effort is still focused on further optimizing WSN technology, e.g., to optimize data communication to minimize energy consumption [123,124]. In addition, sensor information models such as the OGC Sensor Observation Service [125] are being developed to describe WSN data and provide information needed for searching observation locations and sensors [126]. This will support the establishment of cyber–physical infrastructures that provide solutions for the integrated management of heterogeneous data resources (e.g., live sensors, sensor models, and simulation systems) and collaborative observation systems based on multiple platforms (e.g., WSN and remote sensing) to more easily provide researchers with relevant geoscientific information [127].

More recently, the Narrow Band-Internet of Things (NB-IoT) communication technology has been introduced as an emerging technology in the field of WSN [128]. NB-IoT has the potential to revolutionize WSN, as it is deployed jointly with existing mobile cellular networks and, thus, enables large-scale and possibly continental coverage in the near future [129]. Furthermore, the emerging 5G network is expected to further improve communication and its design and standardization will support NB-IoT applications [130]. A wide range of smart applications, including smart farming, are expected to be supported in the near future with high-speed massive connectivity under the same roof of 5G wireless communication [131].

With the ongoing development of wireless sensor networks and advances in computing technology (e.g., edge computing and machine learning), governance issues are becoming increasingly important for the design and application of WSN [132]. The full potential of soil hydrological sensor networks has yet to be realized. Therefore, the focus needs to be broadened from developing better WSN applications and optimizing their technical operation (e.g., procuring more energy-efficient and effective electronic components) to issues arising from pressing environmental problems, such as addressing climate change and food security and providing information for increasingly complex and highly resolved Earth system models [133].

## Figures and Tables

**Figure 1 sensors-22-09792-f001:**
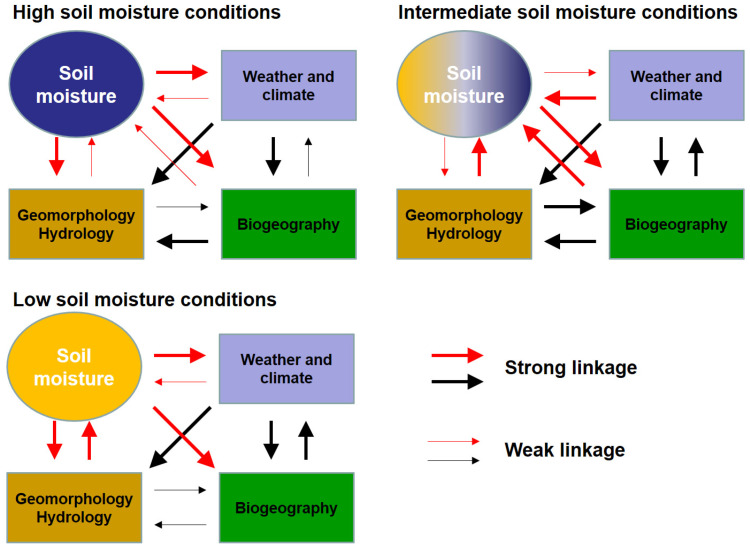
Illustration of linkages among soil moisture, weather and climate, geomorphology, hydrology, and biogeography for high (**upper left**), intermediate (**upper right**), and low (**lower left**) soil moisture conditions (adapted from Legates et al. [22]). The red linkages indicate direct relationships between soil moisture and weather/climate, geomorphology/hydrology, and biogeography. The black arrows indicate linkages independent of soil moisture.

**Figure 2 sensors-22-09792-f002:**
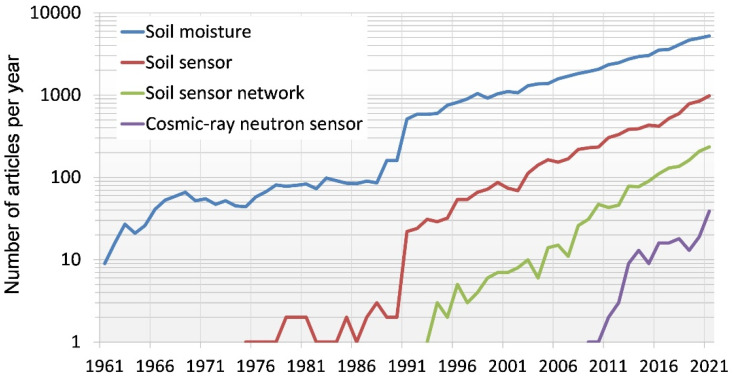
Number of articles on soil moisture, soil sensors, soil sensor networks, and cosmic-ray neutron sensors per year since 1961.

**Figure 3 sensors-22-09792-f003:**
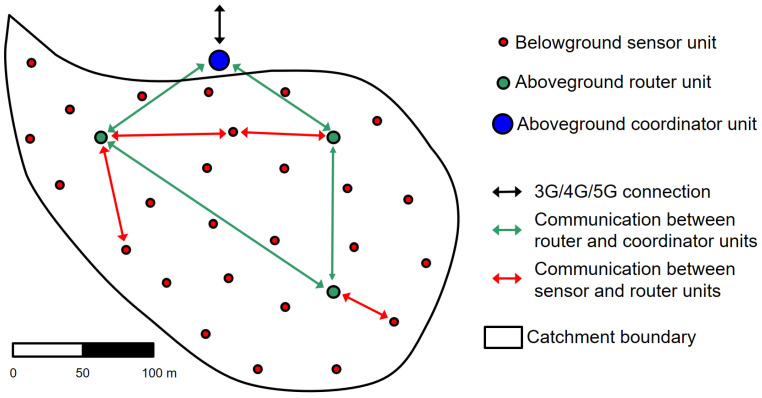
The hybrid wireless underground network topology of SoilNet exemplified for a virtual catchment area (adapted from Bogena et al. [14]).

**Figure 4 sensors-22-09792-f004:**
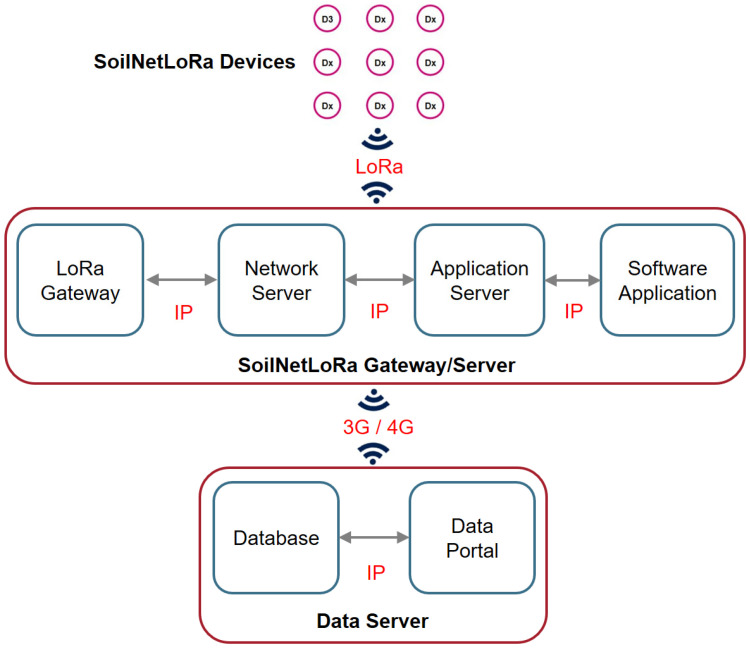
Principle of the LoRa network topology and its basic system architecture, as well as data communication types demonstrated using the example of the SoilNetLoRa wireless sensor network.

**Figure 5 sensors-22-09792-f005:**
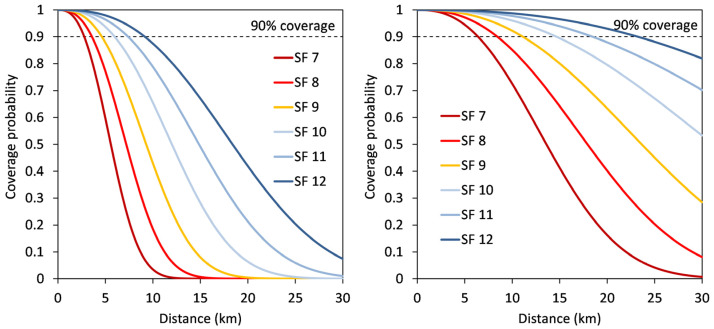
Coverage probabilities for path loss exponents 2.5 (**left**) and 2.7 (**right**) using different spreading factors on a carrier frequency of 868.5 MHz and radio link distances (from 0–30 km). The dashed line depicts the 90% coverage probability.

**Figure 6 sensors-22-09792-f006:**
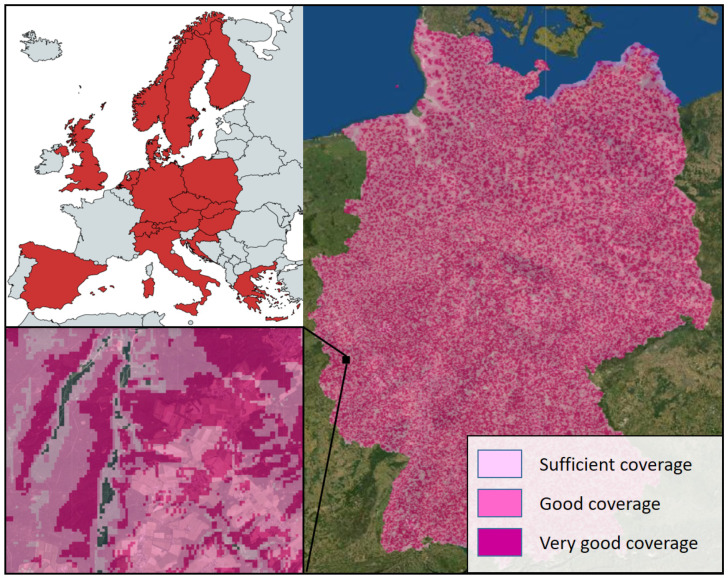
European countries where NB-IoT services are offered (**upper left**), NB-IoT coverage for Germany provided by Deutsche Telekom (**right**), and a large-scale detailed map indicating local NB-IoT gaps in deep valleys in a rural Eifel region in Germany (**lower left**). The NB-IoT coverage maps are based on available interactive maps [55].

**Table 1 sensors-22-09792-t001:** Receiver sensitivities for different spreading factors and the corresponding bit rates. The parameter *q_SF_* indicates the specific noise threshold for a given spreading factor. Values are taken from Georgiou and Raza [48].

Spreading Factor	Sensitivity (dBm)	*q_SF_* (dBm)	Bit Rate (bits/s)
7	−123	−6	5469
8	−126	−9	3125
9	−129	−12	1758
10	−132	−15	977
11	−134.5	−17.5	537
12	−137	−20	293

**Table 2 sensors-22-09792-t002:** Examples of soil moisture sensors commonly used in WSN applications.

Sensor Name	Sensor Type	Manufacturer	WSN Applications
EC-5/EC-TM	Capacitance	METER Group	[14,32,65,66,67,68]
ThetaProbe ML2x	Capacitance	Delta-T Devices	[69]
GS3/TEROS11	Capacitance	METER Group	[70]
TDR probes	TDR	Custom-made	[71]
ACC-SEN-TDT	TDT	Acclima Inc.	[72]
SISOMOP/SPADE	TDT similar	sceme.de GmbH	[16,73,74]
SMT100	TDT similar	Truebner GmbH	[75,76,77,78]

## Data Availability

Not applicable.

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
