# Peer review of "Recent Developments in Wireless Soil Moisture Sensing to Support Scientific Research and Agricultural Management"

_sensors, 2022, doi:10.3390/s22249792_

Round 1

Reviewer 1 Report

This paper is interesting and takes important problem of soil moisture sensing use into scientific research and crop management. Paper structure and source materials are appropriate. The proposed methodology and obtained results are encouraging and allows for need next investigation in different environmental conditions.  Some comments are provided in the text (enclosed pdf).

To discussion

I propose add the some information about the soil moisture sensing along soil profile with relevant references.

Author Response

We thank the reviewer for the comments, which we have addressed in the revised manuscript. Please see the attachment for the point-by-point response to the reviewer’s comments.

Best wishes

Heye Bogena

Reviewer 2 Report

The last reviewers comments and a reduced plagiarism percentage comment are not incorporated in the revised version of the manuscript. However, a statement by the authors is made to do so. Moreover, I do have the following comments for further development of the manuscript.

L50: Change “…can be defined:” to either “…can define:” or to “…can be defined by”

Figure 1: what do colors (red & black) of the arrows refer to?

Figure 2: No need include how data obtained in the figure caption. This detail can be moved to the end of last paragraph of P2.

L117: swap the term to be: “Narrow Band Internet of Things (NB-IoT)”

L119-121: Authors need to sate a clear justification. It is not clear what is the novelty of this review paper. What is gap this study is trying to cover. An overview of soil moisture monitoring using very recent technologies using and WSN and IoT has been covered by many papers including the use of NBIoT.

L149: The paper needs to focus in what is most recently used of tech in this field not the “the most widely” ones.

Section 5. In general, the SoilNet is dominating this section which gives it less weight rather than exploring more applications with recent advanced tech. in this field.

Section 5.2. This section is very brief to the point it misses a lot of recent advanced RS applications in this field. In addition, what about the other RS platforms than satellites?

Author Response

(The authors gave the same response as above.)
